# Post-COVID-19 Syndrome: Audiometric Findings in Patients with Audiological Symptoms

**DOI:** 10.3390/ijerph20176697

**Published:** 2023-09-01

**Authors:** Oscar O. Ríos Coronado, Claudia A. Igual Félix, Gabriel Paz Flores, Magdicarla E. De Alba Márquez, Cynthia R. Cárdenas Contreras, Esteban González Díaz, Ana I. Sedano Paz, Luis R. González-Lucano

**Affiliations:** 1Hospital Civil de Guadalajara, Guadalajara 44340, Mexico; gpaz@hcg.gob.mx (G.P.F.); magdicarla.dealba@academicos.udg.mx (M.E.D.A.M.); ccardenas@hcg.gob.mx (C.R.C.C.); esteban.gdiaz@academicos.udg.mx (E.G.D.); 2Tecnologico de Monterrey, Escuela de Medicina y Ciencias de la Salud, Monterrey 64710, Mexico; a01250598@tec.mx; 3Hospital Regional Instituto de Seguridad y Servicios Sociales de los Trabajadores del Estado Valentín Gómez Farías, Zapopan 45100, Mexico; isabel.sedano6244@alumno.udg.mx

**Keywords:** hearing loss, post-COVID 19 syndrome, audiometric findings, observational, cross-sectional study

## Abstract

Since the SARS-CoV-19 pandemic, the possibility of audiological involvement by this virus has been speculated without being able to generate a true cause–effect relationship. The objective of this observational, descriptive cross-sectional study is to describe the audiometric findings of post-COVID-19 patients with audiological symptoms. A sample of 47 patients with a diagnosis of COVID-19 infection was included: The age range was between 18 and 50 years old, the mean age was 37.0 years with a standard deviation of ±8.3 years, and 32 patients (68.1%) were female and 15 male patients (31.9%). Patients were recruited by the Otolaryngology service at Civil Fray Antonio Mayor Hospital from September 2020 to December 2022. Tonal audiometry was performed in a window of no more than 3 months from the onset of symptoms. The Chi-square test was used and odds ratios (OR) were established to associate the variables of post-COVID-19 audiological symptoms and the prevalence of hearing loss. A 95% confidence interval (CI) and statistical significance were considered of *p* ≤ 0.05. The audiological symptoms presented a prevalence of 74.4% for a sensation of ear fullness, 59.6% for tinnitus, and 51.1% for a sensation of hearing loss.

## 1. Introduction

Hearing loss is a disease considered to be the fourth cause of disability worldwide, since it conditions difficulty in the person who suffers from it not only to develop socially but educationally and even impacts their ability to earn money [1]. As we emerge from the pandemic’s acute phase, it is imperative that we direct our focus toward post-COVID research. Studies have reported a wide range of audiological manifestations in post-COVID-19 patients, with sensorineural hearing loss (SNHL), tinnitus, and/or vertigo being described to occur during and following COVID-19 infection [2]. Audiological research in post-COVID-19 is both novel and significant due to the unique nature of the pandemic, the large number of affected individuals, and the diverse audiological manifestations observed. The objective of this study is to explore potential correlations between audiological symptoms observed during or after COVID-19 and their possible association with audiometric findings. By exploring this connection, we hope to gain a better understanding of the lasting impact that COVID-19 can have on our ability to hear, and lay the groundwork for future reviews. The scale of individuals potentially affected by auditory complications makes this research significant as it addresses a considerable public health concern.

A person is said to have hearing loss when they do not retain or cannot hear as well as a person with normal hearing. Normal hearing is defined when, through an audiometry study, the patient has hearing thresholds of 20 dB or less [3]. Since the emergence of the SARS-CoV-19 pandemic, the possibility of audiological involvement by this virus has been speculated without being able to generate a true cause–effect relationship, documenting mostly case reports in an attempt to describe possible atypical clinical presentations due to COVID-19 infection. Infection by the new coronavirus, SARS-CoV-2, has marked history with a pandemic that reached surprising numbers in infections and deaths. Its clinical manifestations range from cough to severe systemic disease with multiple organ failure. Ear, nose, and throat manifestations have gained importance due to the high frequency, specifically, of anosmia and dysgeusia with high predictive values for infection. Nevertheless, chronic manifestations have yet to be properly explored, post-COVID-19 syndrome—also referred to as long-COVID—is a condition occurring three months after a confirmed or suspected SARS-CoV-2 infection, lasting at least two months, with symptoms that cannot be explained by alternative diagnoses [3] In previous studies, such as a case series of 10 patients with audiovestibular symptoms, including hearing loss, was reported. Of these patients, seven developed audiovestibular symptoms after other symptoms of COVID-19. The audiometry performed showed that all patients experienced sensorineural hearing loss ranging from mild to profound. All patients underwent magnetic resonance imaging during COVID-19 infection to rule out retro cochlear pathology, finding ipsilateral enhancement of the labyrinth, geniculate ganglion, and the facial nerve in one patient, consistent with inflammation related to viral infection, ipsilateral to the ear with sensorineural hearing loss [4]. There is limited research on how audiological involvement can be impaired or diminished even after the acute phase of a viral infection.

Studies have found alterations in hearing thresholds and up to a 15% incidence of audiological symptoms in post-COVID-19 patients without being able to establish a clear relationship [5]. The pathophysiological mechanisms of cell invasion by a viral etiology include the presence of angiotensin-converting enzyme 2 (ACE2) in the inner and middle ear [6], disproportionate proinflammatory cytokines, wherein inflammation derived from NETosis plays an important role in the development of cytokine storm, sepsis, and multiple organ failure, in which surrounding tissue damage and coagulopathy could explain possible effects on the middle and inner ear [7]. The relationship between hearing loss and viral infections has been demonstrated as its etiology by various mechanisms such as direct viral injury, as well as injury due to inflammatory reactions in the sensory organ [8]. Due to the above, it has been hypothesized that COVID-19 could be one more agent in this list of viral etiologies of hearing loss.

The present article seeks to find evidence on hearing threshold alterations provided by audiometry that could be associated with COVID-19 infection presenting audiological symptoms.

## 2. Materials and Methods

The present observational cross-sectional study focuses on reporting audiometric findings of post-COVID-19 patients who report audiological symptoms, during or after the onset of symptoms, performing an audiometry study in a window of no more than 3 months from the onset of symptoms. Patients with a history of otologic disease or consumption of ototoxic drugs during their illness were excluded. Qualitative methodology using an observational, descriptive method, and cross-sectional study via interviewing was used to explore patients with a positive PCR or antigen COVID test presenting audiological symptoms. This study adheres to the ethical principles for medical research in human beings outlined in the Declaration of Helsinki promulgated by the World Medical Association. The present study approached patients diagnosed with SARS-CoV-2 of fewer than 3 months of evolution and treated at the Civil Fray Antonio Mayor Hospital from September 2020 to December 2022. Patients were recruited by personnel from the otorhinolaryngology and head and neck surgery service. The method of selection was non-probabilistic, and all patients who sought attention at the otorhinolaryngology service during the study period were captured. A total of 47 patients with a diagnosis of COVID-19 infection were included, between 18 and 50 years of age. To comply with legal age regulations, a minimum age requirement of 18 was established. In order to avoid potential bias caused by presbycusis or otosclerosis, which commonly occur around age 50, an age limit was set at that threshold. Patients presented audiological symptoms at the beginning, during, or after the COVID-19 infection, performing audiometry in a window of no more than 3 months from the onset of symptoms.

A non-probabilistic, sequential sampling was carried out. There was only one study group determined by the selection criteria. Inclusion criteria were: (a) patients from 18 years to 50 years of age with a positive diagnosis of COVID-19; (b) patients with a positive result for COVID-19 by PCR or antigen test; (c) patients who agreed to be part of the research protocol in question; (d) patients who in the interview referred any of the audiological symptoms from their diagnosis of COVID-19.

Although patients had no previous audiometry, they reported a decrease in their hearing clarity during or after their infection. This was identified through the auditory symptomatology questionnaire. The patients never reported any previous auditory symptoms. Further confirmation of any audiological impairment such as hearing loss was confirmed by tonal audiometry. During the clinical evaluation of patients with COVID-19, they were asked if they had experienced tinnitus prior to their infection or if they developed it during or after their illness. The focus was on direct questioning about the presence of tinnitus and its timing in relation to COVID-19. The degree of disability from tinnitus was not explored during this evaluation.

Exclusion criteria were: (a) patients with a history of unilateral or bilateral conductive or sensory hearing loss before a diagnosis of COVID-19; (b) previous diagnosis of another cochleovestibulopathy; (c) patients with a history of listed otologic surgery: stapedectomy, cochlear implant; (d) patients who have consumed the following ototoxic drugs during their infection: aminoglycosides: amikacin, gentamicin, kanamycin, neomycin, plazomicin, streptomycin, and tobramycin; quinoline derivatives: chloroquine, quinine, and hydroxychloroquine; loop diuretics: furosemide, ethacrynic acid, and bumetanide; and platinum-derived chemotherapeutics.

Elimination criteria were: (a) patients who did not have their audiometry performed within the first 3 months after their diagnosis of COVID-19. This study did not exclude patients who had previously received a COVID-19 vaccine. Out of the 47 patients who were evaluated, 4 (8.5%) did not disclose any prior vaccination for SARS-CoV-2. Meanwhile, 20 (42.5%) reported having completed their vaccination scheme, while 23 patients (48.9%) had an incomplete scheme of their vaccination status.

Forty-seven patients were selected for the study, a total of ninety-four ears, and these patients attended the COVID service department and the Otorhinolaryngology service, both from the “Fray Antonio Alcalde” Civil Hospital from September 2020 to December 2022. Patients were then entered into the database met the inclusion criteria and presented complete data.

The patient was provided with complete information regarding the study protocol, utility, and eligibility was confirmed by a survey of symptoms (Figure 1). Before carrying out the clinical and paraclinical evaluations, the written informed consent was filled out by selected patients and any questions were answered. Findings were then sent to the Audiology Service of the Civil Hospital of Guadalajara “Fray Antonio Alcalde”, Decentralized Public Organization, where the patient was evaluated audio-logically, consisting of otoscopy, speech audiometry, tympanometry, and tonal audiometry to determine the hearing threshold.

First, we explained to the patient what the test consists of and its objective. Secondly, we performed an otoscopy to ensure a permeable external auditory canal. Next, the patient was seated in a soundproof booth and sent a tone produced by the audiometer that reached them through headphones for air conduction and a vibrator for bone conduction, placed in contact with the mastoid process. The order of the frequencies to be studied was the following: 1000, 2000, 4000, 8000, 500, and 250 Hz. The patient had to answer each time they heard the sound, even faintly, by pressing a button, thus detecting the minimum sound stimulus and auditory threshold that the patient perceived at different frequencies, in both air and bone conduction. The tones were not presented rhythmically but randomized to prevent the patient from anticipating the responses. The detected thresholds were recorded using the internationally accepted signs, obtaining a graph of the patient’s hearing known as an audiogram.

Speech audiometry was performed, as part of the audiometry study to evaluate the integrated ability of the subject to detect, recognize, and differentiate phonemes in this case, of the Spanish language.

Hearing loss was identified by decreased sensitivity of the auditory system to a sound stimulus, resulting in a subjective sensation of increased hearing threshold by the patient. The diagnosis was made with the determination of the hearing threshold through tonal audiometry, determined as present or absent [2]. Types of hearing loss were classified based on the result of the audiometry, which made it possible to discern the anatomical site of the possible lesion that produced the hearing impairment [2,9].

I.Conduction: Lesion located at the level of the middle ear or external ear, evidenced in the tonal audiometry with an alteration of the thresholds in the airway compared to normal in the bone pathway [9].II.Sensorineural: Lesion located in any of the areas of the auditory pathway such as cochleopathy (Corti’s organ), neuropathy (acoustic pathway), or corticopathy (auditory cortex), evidenced in the tonal audiometry with an alteration of the tonal thresholds of both the airway and the bone pathway equally [9].III.Mixed: Lesions in the neurosensory pathway (cochlea, neural pathway, and/or auditory cortex) and conductive pathway (middle and/or external ear) coexisting, presenting in the audiometry a combination of previously described patterns of conductive and neurosensory lesion [9].

Using the Shapiro–Wilk test, a non-parametric distribution was established, the Chi-square test was used and odds ratios (OR) were established to associate the variables of post-COVID-19 audiological symptoms and the prevalence of hearing loss. A confidence interval (CI) of 95% and a statistical significance of *p* ≤ 0.05 were considered.

## 3. Results

This study enrolled a total of 47 patients who are from the western region of Mexico. The Civil Fray Antonio Mayor Hospital, one of the largest hospitals in the country, receives a significant number of patients on a daily basis. These patients attended the COVID service and the otorhinolaryngology service, from September 2020 to December 2022, in Mexico. Patients were entered into the database, met the inclusion and exclusion criteria, and presented complete and full data. Out of the total number of patients evaluated, the presence of comorbidities was assessed. Among them, 37 (78.7%) did not suffer from any comorbidities. Meanwhile, 3 (6.4%) had a smoking history, 2 (4.3%) had systemic arterial hypertension, 1 (2.1%) had type II diabetes mellitus, 1 (2.1%) had type I diabetes, 1 (2.1%) had depression, 1 (2.1%) had depression and anxiety, and 1 (2.1%) had both smoking and hypothyroidism.

Hearing loss was identified by decreased sensitivity of the auditory system to a sound stimulus, resulting in a subjective sensation of increased hearing threshold by the patient. The diagnosis was made with the determination of the hearing threshold through tonal audiometry, determined as present or absent [3]. A total of 18 patients (38%) with some degree of hearing loss was reported, of which 4 (8.5%) experienced sudden hearing loss. Audiological symptoms presented a prevalence of 74.4% for a sensation of ear fullness, 59.6% for tinnitus, 51.1% for a sensation of hearing loss, and 36.2% for the need to repeat words to understand conversations. Of the above, the sensation of hearing loss was associated with an OR = 2.94 (*p* = 0.033) and the need to repeat words with an OR = 3.802 (*p* = 0.008) to some degree of hearing loss in the audiometry.

The final sample consisted of 47 patients, and audiological symptoms referred by patients were evaluated. First, regarding the presence or absence of tinnitus, 19 patients (40.4%) did not refer tinnitus; 28 patients (59.6%) reported the presence of tinnitus during and/or after COVID-19, of which 10 (21.3%) patients reported left tinnitus, 10 (21.3%) right tinnitus, and 8 (17%) bilateral tinnitus (Figure 2).

Regarding tinnitus, 59.6% of the patients presented this audiological symptom. The relationship of tinnitus (37% of the 94 ears evaluated) with the presence of hearing loss in the tonal audiometry presented an OR of 1.960 (0.721–5.329, 95% CI; *p* = 0.183) being insufficient to establish a relationship between these two variables.

Secondly, from the final sample (47 patients) evaluated, 23 patients (48.9%) did not report a sensation of hearing loss; 24 patients (51.1%) did refer it, of which 5 (10.6%) patients reported a sensation of hearing loss in the left ear, 6 (12.8%) a sensation of hearing loss in the right ear, and 13 (27.7%) a sensation of bilateral hearing loss (Figure 3).

The sensation of hearing loss during or after COVID-19 was reported in 39% of the 94 ears studied (51.1% of patients); presenting 2.94 times more risk of presenting a diagnosis of hearing loss in audiometry when presenting said symptom, given by OR = 2.940 (1.064–8.121, 95% CI; *p* = 0.033).

Thus, establishing a statistically significant relationship. Four of the patients evaluated (8.5%) met diagnostic criteria for unilateral sudden hearing loss, establishing an incidence of sudden hearing loss in post-COVID-19 patients with audiological symptoms.

The necessity of word repeating during a conversation was found in 17 patients (36.2%), the relationship between this symptom and the presence of hearing loss in audiometry presented an OR = 3.802 (1.359–10.636, 95% CI; *p* = 0.008). This is significant enough to establish a relationship, and strongly suggests the possibility of this symptom is a potential risk factor for hearing loss in post-COVID-19 patients (Figure 4).

Audio metrical correlations were established and a total of 18 patients were affected by some degree of unilateral or bilateral hearing loss (38% prevalence). According to the classification of the World Health Organization [10], the degree of hearing loss was determined, according to the tonal threshold (by PTA of 3 frequencies), as follows:

Normal hearing, ≤20 dB.

Mild hearing loss, 20 to <35 dB.

Moderate hearing loss, 35 to <50 dB. Moderate to severe hearing loss, 50 to <65 dB.

Severe hearing loss, 65 to <80 dB.

Profound hearing loss, 80 to <95 dB.

Anacusis, 95 dB or higher.

Of the 94 ears, 74 (78.72%) had normal hearing, from which 70 (74.5%) ears were without underlying pathology, and 4 ears (4.3%) presented ototubaritis. In total, 8 (8.5%) patients had mild conductive hearing loss, 7 (7.4%) mild conductive hearing loss and ototubaritis, 12 (12.76%) patients presented sensorineural hearing loss, 1 (1.1%) had mild sensorineural hearing loss, 5 (5.3%) had moderate sensorineural hearing loss, 4 (4.3%) presented sudden sensorineural hearing loss (1 profound, 1 severe, 2 anacusis), and 1 (1.1%) mild sensorineural hearing loss and ototubaritis, 1 (1.1%) moderate mixed hearing loss.

## 4. Discussion

This study explored persistent audiological symptoms regarding post-COVID-19 syndrome and audio-metrical involvement. Various studies have shown how feasible the infection of the inner ear by COVID-19 is, as well as its possible involvement not only directly but also due to inflammatory “crossfire” or vascular causes that could culminate in audiovestibular affectations [11,12]. The present study included patients who, during or after their COVID-19 infection, presented tinnitus, ear fullness, and a sensation of hearing loss, integrating 47 patients, 32 (68.1%) women, and 15 (31.9%) men, with results in the audiometry of some degree of hearing loss in almost one fifth (19%) of the 94 ears studied. Studies have revealed that viral infections can be a significant factor in hearing loss, as they can cause direct damage and inflammation in the sensory organs. As such, it is important to consider the possibility of COVID-19 being a potential viral cause of hearing loss.

The presence of tinnitus is considered one of the most prevalent symptoms in the general population (12 to 30%) and various etiologies are attributed to it, such as upper respiratory tract diseases, hearing loss, or mood factors such as stress and depression [13]. Even children can experience audiological symptoms like tinnitus, aural fullness, and hearing loss due to COVID-19. This occurs in a significant proportion of the pediatric population [14]. Experiencing hearing loss during or after contracting COVID-19 has been found to have a significant correlation with a diagnosis of hearing loss in audiometry. This highlights the importance of investigating this clinical manifestation during the early stages of infection. It is recommended to include hearing assessments as part of the primary audiological first contact assessment to gain insight into potential complications caused by the virus.

Jafari et al. in their systematic review and meta-analysis describe that COVID-19 could cause hearing loss; however, most of the studies that try to establish a relationship between coronavirus and sudden hearing loss are case reports, so care should be taken in result interpretation [15]. According to Spinato et al., by focusing on areas with fatal and fatal outcomes such as severe pneumonia, sepsis, and septic shock, symptoms or data that do not immediately put the patient’s life at risk, such as those of the upper airways with otorhinolaryngological manifestations, were left behind by 69% of those infected, resulting in an important cause of disability potentially unattended in the population [16].

The necessity of word repeating as an audiological symptom presented a correlation for the development of hearing loss in audiometry and strongly suggests the possibility of this symptom being a potential risk factor for hearing loss in post-COVID-19 patients. Therefore, it is important to specifically ask this question, along with the sensation of hearing loss in the interrogation of patients evaluated for COVID-19, in the event that the patient refers to these two symptoms, and to be able to perform audiometry in a timely manner to rehabilitate the patient if necessary and thus reduce the impact on their quality of life, and professional, social, and educational development.

### Limitations

In light of the then-ongoing pandemic and the highly contagious nature of the disease, some patients had expressed reservations about attending their scheduled audiometry appointments, despite having been informed of the extensive precautions in place to ensure a safe and sanitized environment. It is worth noting that any failure of the otolaryngology service’s audiometry equipment could have potentially impeded the progress and recruitment of patients.

## 5. Conclusions

The prevalence of hearing loss in post-COVID-19 patients with audiological symptoms is 38% in this study. The degree of hearing loss and type of hearing loss in post-COVID-19 patients with audiological symptoms ranges from mild to predominantly sensorineural anacusis as the degree of hearing loss increases. The prevalence of sudden sensorineural hearing loss in post-COVID-19 patients with audiological symptoms in this study is 8.5%. The presence of a sensation of hearing loss is related to a risk factor for presenting some degree of hearing loss in tonal audiometry in post-COVID-19 patients with audiological symptoms, along with a strong correlation with the necessity of word repeating during a conversation. Investigating these diverse audiological manifestations is crucial for a comprehensive understanding of the virus’s impact on the auditory system. Identifying and addressing auditory issues in post-COVID-19 patients can lead to the development of targeted rehabilitation strategies, improving their overall functional outcomes and emotional well-being. Early identification and intervention for audiological issues in post-COVID-19 patients can lead to better outcomes. Audiological research can help identify risk factors and symptoms associated with auditory complications, enabling healthcare providers to initiate timely interventions and reduce the burden of long-term auditory problems. This is the initial stage of comprehensive research that will thoroughly analyze variables researched within this study such as tinnitus, over a prolonged period.

## Figures and Tables

**Figure 1 ijerph-20-06697-f001:**
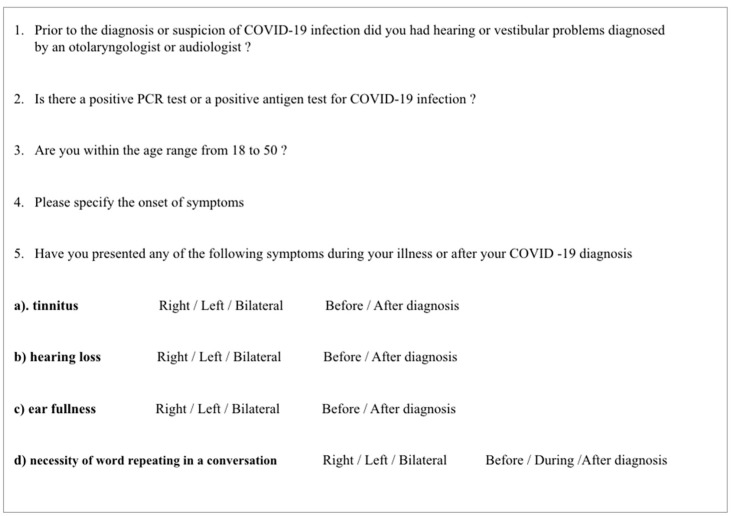
COVID-19 and audiological symptoms survey.

**Figure 2 ijerph-20-06697-f002:**
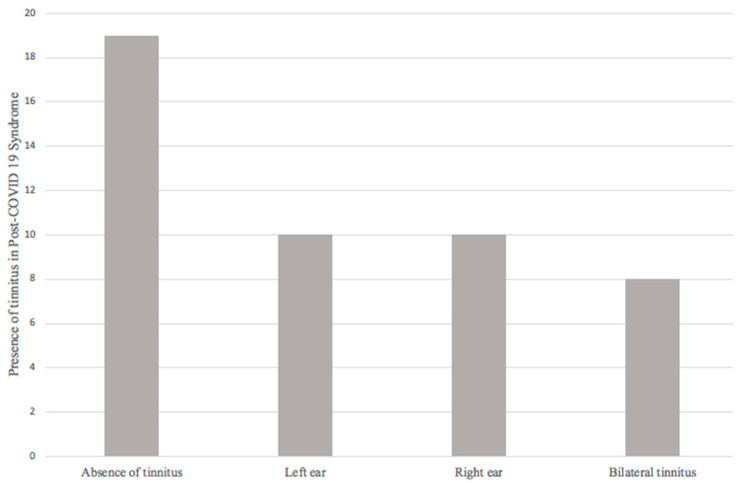
Presence of tinnitus in post-COVID-19 syndrome.

**Figure 3 ijerph-20-06697-f003:**
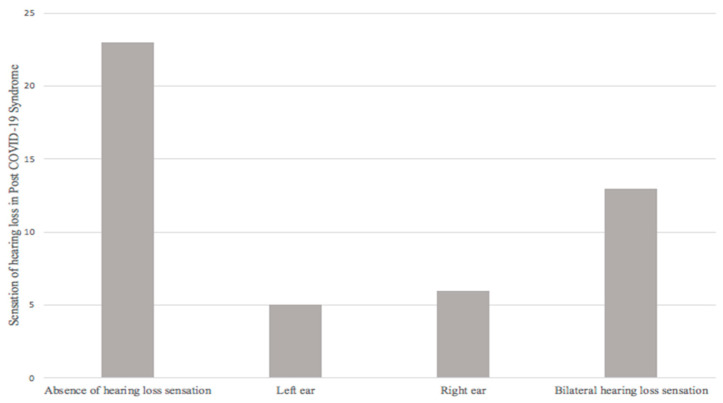
Sensation of hearing loss in post-COVID-19 syndrome.

**Figure 4 ijerph-20-06697-f004:**
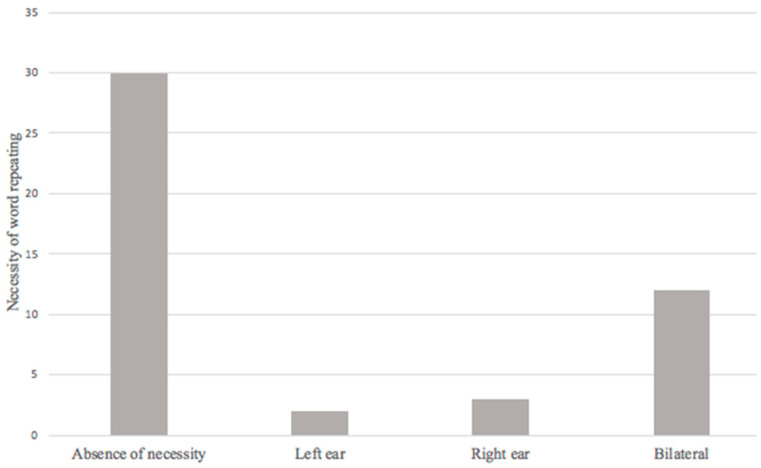
Necessity of word repeating in post-COVID-19 syndrome.

## Data Availability

Study data are available upon reasonable request to the corresponding author.

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
