# Peer review of "Post-COVID-19 Syndrome: Audiometric Findings in Patients with Audiological Symptoms"

_ijerph, 2023, doi:10.3390/ijerph20176697_

Round 1

Reviewer 1 Report

I would like to thank the authors for their submission and allowing me to review their work.

This is an interesting study on an important topic. However, I would be grateful if you could add further explanations and changes on the following points:

1) ABSTRACT: Page 1, line 13

Please specify the mean age (± standard deviation) and gender of the study population.

2) ABSTRACT: Page 1, line 16

Where and when was the study conducted? Please specify the methods of the study.

3) MATERIALS AND METHODS: Page 2, line 95

Were patients with genetic syndromes (potentially associated with hearing loss), recent head trauma or a family history of hearing loss excluded from the study?

4) MATERIALS AND METHODS: Page 2, line 95

Were vaccinated patients excluded from the study?

5) MATERIALS AND METHODS: Page 4, line 154

How tinnitus was investigated? Did you use specific questionnaires such as Tinnitus Handicap Inventory (THI)?

6) RESULTS: Page 4, line 162

Please specify in which country the study was conducted (Mexico).

7) RESULTS: Page 4, line 166

Please add more specific demographic and clinical information about the study participants.

8) DISCUSSION: Page 7, line 255

In the discussion section, I suggest adding that COVID-19 can cause audiological symptoms (such as tinnitus, aural fullness, hearing loss) even in a high percentage of the pediatric population.

Reference: Audiological and vestibular symptoms following SARS-CoV-2 infection and COVID-19 vaccination in children aged 5-11 years. Am J Otolaryngol. 2023;44(1):103669. doi:10.1016/j.amjoto.2022.103669)

9) DISCUSSION: Page 8, line 265

A “limitations section” should be added (e.g. the study is not population based, limited number of patients, single center…)

10) DISCUSSION: Page 8, line 265

Which are the future prospects of this study?

Minor editing of English language required

Author Response

Thank you for taking the time to review our article. We value your input and are committed to addressing the points raised in a constructive manner to further improve the quality and accuracy of our work.

Author Response

Thank you for your valuable insights on the report. 
